# A Unified Algebraic Perspective on Lipschitz Neural Networks

**Alexandre Araujo**[*1]**, Aaron Havens**[*2]**, Blaise Delattre**[3,4]**, Alexandre Allauzen**[3,5] **and Bin Hu**[2]

[1] INRIA, Ecole Normale Supérieure, CNRS, PSL University, Paris, France
[2] CSL & ECE, University of Illinois Urbana-Champaign, IL, USA
[3] Miles Team, LAMSADE, Université Paris-Dauphine, PSL University, Paris, France
[4] Foxstream, Vaulx-en-Velin, France
[5] ESPCI PSL, Paris, France

## Abstract

Important research efforts have focused on the design and training of neural networks with a controlled Lipschitz constant. The goal is to increase and sometimes guarantee the robustness against adversarial attacks. Recent promising techniques draw inspirations from different backgrounds to design 1-Lipschitz neural networks, just to name a few: convex potential layers derive from the discretization of continuous dynamical systems, Almost-Orthogonal-Layer proposes a tailored method for matrix rescaling. However, it is today important to consider the recent and promising contributions in the field under a common theoretical lens to better design new and improved layers. This paper introduces a novel algebraic perspective unifying various types of 1-Lipschitz neural networks, including the ones previously mentioned, along with methods based on orthogonality and spectral methods. Interestingly, we show that many existing techniques can be derived and generalized via finding analytical solutions of a common semidefinite programming (SDP) condition. We also prove that AOL biases the scaled weight to the ones which are close to the set of orthogonal matrices in a certain mathematical manner. Moreover, our algebraic condition, combined with the Gershgorin circle theorem, readily leads to new and diverse parameterizations for 1-Lipschitz network layers. Our approach, called SDP-based Lipschitz Layers (SLL), allows us to design non-trivial yet efficient generalization of convex potential layers. Finally, the comprehensive set of experiments on image classification shows that SLLs outperform previous approaches on certified robust accuracy. Code is available at `github.com/araujoalexandre/Lipschitz-SLL-Networks`.

## 1 Introduction

The robustness of deep neural networks is nowadays a great challenge to establish confidence in their decisions for real-life applications. Addressing this challenge requires guarantees on the stability of the prediction, with respect to adversarial attacks. In this context, the Lipschitz constant of neural networks is a key property at the core of many recent advances. Along with the margin of the classifier, this property allows us to certify the robustness against worst-case adversarial perturbations. This certification is based on a sphere of stability within which the decision remains the same for any perturbation inside the sphere (Tsuzuku et al., 2018).

The design of 1-Lipschitz layers provides a successful approach to enforce this property for the whole neural network. For this purpose, many different techniques have been devised such as spectral normalization (Miyato et al., 2018; Farnia et al., 2019), orthogonal parameterization (Trockman et al., 2021; Li et al., 2019; Singla et al., 2021; Yu et al., 2022; Xu et al., 2022), Convex Potential Layers (CPL) (Meunier et al., 2022), and Almost-Orthogonal-Layers (AOL) (Prach et al., 2022). While all these techniques share the same goal, their motivations, and derivations can greatly differ,

---

* Equal contribution.

delivering different solutions. Nevertheless, their raw experimental comparison fails to really gain insight into their peculiar performance, soundness, and in the end their possible complementarity. Therefore a question acts as a barrier for an in-depth analysis and future development:

***Are there common principles underlying the developments of 1-Lipschitz Layers?***

In this paper, we propose a novel perspective to answer this question based on a unified Semidefinite Programming (SDP) approach. We introduce a common algebraic condition underlying various types of methods like spectral normalization, orthogonality-based methods, AOL, and CPL. Our key insight is that this condition can be formulated as a unifying and simple SDP problem, and that the development of 1-Lipschitz architectures systematically arise by finding "analytical solutions" of this SDP. Our main contributions are summarized as follows.

- We provide a unifying algebraic perspective for 1-Lipschitz network layers by showing that existing techniques such as spectral normalization, orthogonal parameterization, AOL, and CPL can all be recast as a solution of the same simple SDP condition (Theorem 1 and related discussions). Consequently, any new analytical solutions of our proposed SDP condition will immediately lead to new 1-Lipschitz network structures.
- Built upon the above algebraic viewpoint, we give a rigorous mathematical interpretation for AOL explaining how this method promotes "almost orthogonality" in training (Theorem 2).
- Based on our SDPs, a new family of 1-Lipschitz network structures termed as SDP-based Lipschitz layers (SLL) has been developed. Specifically, we apply the Gershgorin circle theorem to obtain some new SDP solutions, leading to non-trivial extensions of CPL (Theorem 3). We derive new SDP conditions to characterize SLL in a very general form (Theorem 4).
- Finally, we show, by a comprehensive set of experiments, that our new SDP-based Lipschitz layers outperform previous approaches on certified robust accuracy.

Our work is inspired by Fazlyab et al. (2019) that develops SDP conditions for numerical estimation of Lipschitz constants of given neural networks. A main difference is that we focus on "analytical SDP solutions" which can be used to characterize 1-Lipschitz network structures.

## 2 RELATED WORK

In recent years, certified methods have been central to the development of trustworthy machine learning and especially for deep learning. *Randomized Smoothing* (Cohen et al., 2019; Salman et al., 2019) is one of the first defenses to offer provable robustness guarantees. The method simply extends a given classifier by the smart introduction of random noise to enhance the robustness of the classifier. Although this method offers an interesting level of certified robustness, it suffers from important downsides such as the high computational cost of inference and some impossibility results from information-theory perspective (Yang et al., 2020; Kumar et al., 2020).

Another approach to certify the robustness of a classifier is to control its Lipschitz constant (Hein et al., 2017; Tsuzuku et al., 2018). The main idea is to derive a certified radius in the feature space by upper bounding the margin of the classifier. See Proposition 1 of Tsuzuku et al. (2018) for more details. This radius, along with the Lipschitz constant of the network can certify the robustness. In order to reduce the Lipschitz constant and have a non-trivial certified accuracy, Tsuzuku et al. (2018) and Leino et al. (2021) both upper bound the margin via computing a bound on the global Lipschitz constant, however, these bounds have proved to be loose. Instead of upper bounding the global Lipschitz constant, Huang et al. (2021b) leverages *local* information to get tighter bound on the Lipschitz constant. On the other hand, other works, instead of upper bounding the local or global Lipschitz, devised neural networks architecture that are provably 1-Lipschitz. One of the first approaches in this direction consists of normalizing each layer with its spectral norm (Miyato et al., 2018; Farnia et al., 2019). Each layer is, by construction, 1-Lipschitz. Later, a body of research replaces the normalized weight matrix by an orthogonal matrix. It improves upon the spectral normalization method by adding the gradient preservation (Li et al., 2019; Trockman et al., 2021; Singla et al., 2021; Yu et al., 2022; Xu et al., 2022). These methods constrain the parameters by orthogonality during training. Specifically, the Cayley transform can be used to constrain the weights (Trockman et al., 2021) and, in a similar fashion, SOC (Singla et al., 2021) parameterizes their layers with the exponential of a skew symmetric matrix making it orthogonal. To reduce cost,

Trockman et al. (2021), Yu et al. (2022), and Xu et al. (2022) orthogonalize their convolutional kernel in the Fourier domain.

More recently, a work by Meunier et al. (2022) has studied Lipschitz networks from a dynamical system perspective. Starting from the continuous view of a residual network, they showed that the parameterization with the Cayley transform (Trockman et al., 2021) and SOC (Singla et al., 2021) correspond respectively to two specific discretization schemes of the continuous flow. Furthermore, a new layer is derived from convex potential flows to ensure the 1-Lipschitz property[1]:

$$z = x - \frac{2}{\|W\|_2^2} W\sigma(W^\top x + b), \tag{1}$$

where $\|W\|_2$ is the spectral norm of the weight matrix $W$ and $\sigma$ is the ReLU activation function. In general, the training of orthogonal layers can be expensive. The Cayley approach involves a matrix inversion, and the implementation of SOC requires either an SVD or an iterative Taylor expansion. The CPL approach can be more efficient, although the computation of $\|W\|_2$ is still needed.

A recent work, *Almost-Orthogonal-layer* (AOL) (Prach et al., 2022) came up with a middle ground: a new normalization which makes the layer 1-Lipschitz by favoring orthogonality. The fully-connected AOL layer is defined as $z = WDx + b$ where $D$ is a diagonal matrix given by[2]:

$$D = \mathrm{diag}\left(\sum_j |W^\top W|_{ij}\right)^{-\frac{1}{2}} \tag{2}$$

They demonstrated that this layer is 1-Lipschitz and they empirically show that, after training, the Jacobian of the layer (with respect to $x$) is almost orthogonal, hence facilitating the training.

Another source of inspiration is the application of convex programs for robustness certification of neural networks (Wong et al., 2018; Raghunathan et al., 2018; Fazlyab et al., 2019; Revay et al., 2020; Fazlyab et al., 2020; Wang et al., 2022). The most relevant work is Fazlyab et al. (2019), which leverages the quadratic constraint approach from control theory (Megretski et al., 1997) to formulate SDPs for estimating the global Lipschitz constant of neural networks numerically. It is possible to solve such SDPs numerically for training relatively small Lipschitz networks (Pauli et al., 2021). However, due to the restrictions of existing SDP solvers, scalability has been one issue when deploying such approaches to deep learning problems with large data sets. Our focus is on the design of Lipschitz network structures, and we avoid the scalability issue via solving SDPs analytically.

## 3  BACKGROUND

**Notation.** The $n \times n$ identity matrix and the $n \times n$ zero matrix are denoted as $I_n$ and $0_n$, respectively. The subscripts will be omitted when the dimension is clear from the context. When a matrix $P$ is negative semidefinite (definite), we will use the notation $P \preceq (\prec)0$. When a matrix $P$ is positive semidefinite (definite), we will use the notation $P \succeq (\succ)0$. Let $e_i$ denote the vector whose $i$-entry is 1 and all other entries are 0. Given a collection of scalars $\{a_i\}_{i=1}^n$, we use the notation $\mathrm{diag}(a_i)$ to denote the $n \times n$ diagonal matrix whose $(i,i)$-th entry is $a_i$. For a matrix $A$, the following notations $A^\top$, $\|A\|_2$, $\mathrm{tr}(A)$, $\sigma_{\min}(A)$, $\|A\|_F$, and $\rho(A)$ stand for its transpose, largest singular value, trace, smallest singular value, Frobenius norm, and spectral radius, respectively.

**Lipschitz functions.** A function $f : \mathbb{R}^n \to \mathbb{R}^m$ is $L$-Lipschitz with respect to the $\ell_2$ norm iff it satisfies $\|f(x) - f(y)\| \leq L\|x - y\|$ for all $x, y \in \mathbb{R}^n$, where $\|\cdot\|$ stands for the $\ell_2$ norm. An important fact is that the robustness of a neural network can be certified based on its Lipschitz constant (Tsuzuku et al., 2018). In this paper, we are interested in the case where $L = 1$. Specifically, we consider the training of 1-Lipschitz neural networks. If each layer of a neural network is 1-Lipschitz, then the entire neural network is also 1-Lipschitz. The Lipschitz constant also satisfies the triangle inequality, and hence convex combination will preserve the 1-Lipschitz property.

**Matrix cones: Positive semidefiniteness and diagonal dominance.** Let $\mathbf{S}^n$ denote the set of all $n \times n$ real symmetric matrices. Let $\mathbf{S}_+^n \subset \mathbf{S}^n$ be the set of all $n \times n$ symmetric positive semidefinite

---

[1]We reverse the transposition from the original layer to have a consistent notation in the rest of the article.

[2]For simplicity, we assume all the columns of $W$ have at least one non-zero entry. Then (2) is well defined.

matrices. It is well known that $\mathbf{S}_+^n$ is a closed-pointed convex cone in $\mathbf{S}^n$. With the trace inner product, $\mathbf{S}_+^n$ is also self-dual. Consider two symmetric matrices $A$ and $B$ such that $A \succeq B \in \mathbf{S}^n$, then we have $A - B \in \mathbf{S}_+^n$, and $\mathrm{tr}(A - B)$ provides a distance measure between $A$ and $B$. In addition, we have $\|A - B\|_F \leq \mathrm{tr}(A - B)$. Finally, the set of all $n \times n$ real symmetric diagonally dominant matrices with non-negative diagonal entries is represented by $\mathbf{D}^n$. It is known that $\mathbf{D}^n$ forms a closed, pointed, full cone (Barker et al., 1975). Based on the Gershgorin circle theorem (Horn et al., 2012), we know $\mathbf{D}^n \subset \mathbf{S}_+^n$. It is also known that $\mathbf{D}^n$ is smaller than $\mathbf{S}_+^n$ (Barker et al., 1975). For any $A \in \mathbf{D}^n$, we have $A_{ii} \geq \sum_{j:j\neq i} |A_{ij}|$. It is important to require $A_{ii} \geq 0$, and the set of real symmetric diagonally dominant matrices is not a cone by itself.

# 4    AN ALGEBRAIC UNIFICATION OF 1-LIPSCHITZ LAYERS

In this section, we present a unified algebraic perspective for various 1-Lipschitz layers (Spectral Normalization, Orthogonalization, AOL, and CPL) via developing a common SDP condition characterizing the Lipschitz property. Built upon our algebraic viewpoint, we also present a new mathematical interpretation explaining how AOL promotes orthogonality in training.

## 4.1    THE UNIFYING ALGEBRAIC CONDITION

First, we present an algebraic condition which can be used to unify the developments of existing techniques such as SN, AOL, and CPL. Our main theorem is formalized below.

**Theorem 1.** *For any weight matrix $W \in \mathbb{R}^{m \times n}$, if there exists a nonsingular diagonal matrix $T$ such that $W^\mathsf{T} W - T \preceq 0$, then the two following statements hold true.*

1. *The mapping $g(x) = WT^{-\frac{1}{2}}x + b$ is 1-Lipschitz.*

2. *The mapping $h(x) = x - 2WT^{-1}\sigma(W^\mathsf{T}x+b)$ is 1-Lipschitz if $\sigma$ is ReLU, $\tanh$ or sigmoid.*

The proof of the above theorem and some related control-theoretic interpretations are provided in the appendix. This theorem allows us to design different 1-Lipschitz layers just with various choices of $T$, in two important cases: for a linear transformation with Statement 1, as well as for a residual and non-linear block with Statement 2. Moreover, for any given weight matrix $W$, the condition $W^\mathsf{T} W \preceq T$ is linear in $T$, and hence can be viewed as an SDP condition with decision variable $T$. To emphasize the significance of this theorem, we propose to derive existing methods used for designing 1-Lipschitz layers by choosing specific $T$ for the SDP condition $W^\mathsf{T} W \preceq T$. The 1-Lipschitz property is then automatically obtained.

- Spectral Normalization (SN) corresponds to an almost trivial choice if we notice that $W^\mathsf{T} W \preceq \|W^\mathsf{T} W\|_2 I \preceq \|W\|_2^2 I$. Hence with $T = \|W\|_2^2 I$, we build the SN layer $g(x) = WT^{-\frac{1}{2}}x + b = \frac{1}{\|W\|_2}Wx + b$.

- The Orthogonality-based parameterization is obtained by setting $T = I$ and enforcing the equality $W^\mathsf{T} W = T = I$. Then obviously $g(x) = Wx + b$ is 1-Lipschitz.

- AOL formula can be derived by letting $T = \mathrm{diag}(\sum_{j=1}^n |W^\mathsf{T} W|_{ij})$. With this choice, we have $T - W^\mathsf{T} W \in \mathbf{D}^n \subset \mathbf{S}_+^n$, hence $W^\mathsf{T} W \preceq T$. Then Statement 1 in Theorem 1 implies that the AOL layer, written as $g(x) = WT^{-\frac{1}{2}}x + b$, is 1-Lipschitz.[3]

- CPL follows the same SN choice $T = \|W\|_2^2 I$, but with Statement 2 of Theorem 1. Hence we derive a different function $h(x) = x - \frac{2}{\|W\|_2^2}W\sigma(W^\mathsf{T}x + b)$ which is also 1-Lipschitz.

The above discussion illustrates the benefit of expressing all these methods within the same theoretical framework, offering us a new tool to characterize the similarity between different methods. For instance, SN and CPL share the same choice of $T = \|W\|_2^2 I$. The difference between them is which statement is used. Hence CPL can be viewed as the "residual version" of

---

[3]For ease of exposition, our main paper always assumes that all the columns of $W$ have at least one non-zero entry such that (2) is well defined. To drop this assumption, we can use a variant of Theorem 1 which replaces the SDP condition with a bilinear matrix inequality condition. We will discuss this point in the appendix.

SN. Clearly, the residual network structure allows CPL to address the gradient vanishing issue more efficiently than SN. With the same approach, we can readily infer from our unified algebraic condition what are the "residual" counterparts for orthogonality-based parameterization and AOL. For orthogonality-based parameterization, if we enforce $W^\mathsf{T}W = T = I$ via methods such as SOC and ECO, then the function $h(x) = x - 2W\sigma(W^\mathsf{T}x + b)$ is 1-Lipschitz (by Statement 2 in Theorem 1). Finally, if we choose $T = \mathrm{diag}\left(\sum_{j=1}^n |W^\mathsf{T}W|_{ij}\right)$, then the function $h(x) = x - 2W\,\mathrm{diag}\left(\sum_{j=1}^n |W^\mathsf{T}W|_{ij}\right)^{-1}\sigma(W^\mathsf{T}x + b)$ is also 1-Lipschitz. Therefore it is straightforward to create new classes of 1-Lipschitz network structures from existing ones.

Another important consequence of Theorem 1 is about new layer development. Any new nonsingular diagonal solution $T$ for the SDP condition $W^\mathsf{T}W - T \preceq 0$ immediately leads to new 1-Lipschitz network structures in the form of $g(x) = WT^{-\frac{1}{2}}x + b$ or $h(x) = x - 2WT^{-1}\sigma(W^\mathsf{T}x + b)$. Therefore, the developments of 1-Lipschitz network structures can be reformulated as finding analytical solutions of the matrix inequality $W^\mathsf{T}W \preceq T$ with nonsingular diagonal $T$. As a matter of fact, the Gershgorin circle theorem can help to improve the existing choices of $T$ in a systematic way. In Section 5, we will discuss such new choices of $T$ and related applications to improve CPL. At this point, it is worth noticing that to develop deep Lipschitz networks, it is important to have analytical formulas of $T$. The analytical formula of $T$ will enable a fast computation of $WT^{-\frac{1}{2}}$ or $WT^{-1}$.

Theorem 1 is powerful in building a connection between 1-Lipschitz network layers and the algebraic condition $W^\mathsf{T}W \preceq T$. Next, we will look closer at this algebraic condition and provide a new mathematical interpretation explaining how AOL generates "almost orthogonal" weights.

**Remark 1.** *The proof of Statement 2 in Theorem 1 relies on (Fazlyab et al., 2019, Lemma 1), which requires the activation function $\sigma$ to be slope-restricted on $[0, 1]$. Therefore, Statement 2 cannot be applied to the case with $\sigma$ being the GroupSort activation function (Anil et al., 2019). In contrast, Statement 1 can be used to build neural networks with any activation functions which are 1-Lipschitz.*

### 4.2 A New Mathematical Interpretation for AOL

In Prach et al. (2022), it is observed that AOL can learn "almost orthogonal" weights and hence overcome the gradient vanishing issue. As a matter of fact, the choice of $T$ used in AOL is optimal in a specific mathematical sense as formalized with the next theorem.

**Theorem 2.** *Given any $W \in \mathbb{R}^{m \times n}$ which does not have zero columns, define the set $\mathbf{T} = \left\{ T : T \text{ is nonsingular diagonal, and } T - W^\mathsf{T}W \in \mathbf{D}^n \right\}$. Then the choice of $T$ for the AOL method actually satisfies*

$$T = \mathrm{diag}(\sum_{j=1}^n |W^\mathsf{T}W|_{ij}) = \arg\min_{T \in \mathbf{T}} \mathrm{tr}(I - T^{-\frac{1}{2}}W^\mathsf{T}WT^{-\frac{1}{2}}) = \arg\min_{T \in \mathbf{T}} \|T^{-\frac{1}{2}}W^\mathsf{T}WT^{-\frac{1}{2}} - I\|_F.$$

We defer the proof for the above result to the appendix. Here we provide some interpretations for the above result. Obviously, the quantity $\|T^{-\frac{1}{2}}W^\mathsf{T}WT^{-\frac{1}{2}} - I\|_F$ provides a measure for the distance between the scaled weight matrix $WT^{-\frac{1}{2}}$ and the set of $n \times n$ orthogonal matrices. If $\|T^{-\frac{1}{2}}W^\mathsf{T}WT^{-\frac{1}{2}} - I\|_F = 0$, then the scaled weight $WT^{-\frac{1}{2}}$ is orthogonal. If $\|T^{-\frac{1}{2}}W^\mathsf{T}WT^{-\frac{1}{2}} - I\|_F$ is small, it means that $WT^{-\frac{1}{2}}$ is "almost orthogonal" and close to the set of orthogonal matrices. Since we require $W^\mathsf{T}W - T \preceq 0$, we know that $I - T^{-\frac{1}{2}}W^\mathsf{T}WT^{-\frac{1}{2}}$ is a positive semidefinite matrix, and its trace provides an alternative metric quantifying the distance between $WT^{-\frac{1}{2}}$ and the set of orthogonal matrices. Importantly, we have the following inequality:

$$\|T^{-\frac{1}{2}}W^\mathsf{T}WT^{-\frac{1}{2}} - I\|_F \leq \mathrm{tr}(I - T^{-\frac{1}{2}}W^\mathsf{T}WT^{-\frac{1}{2}}).$$

If $\mathrm{tr}(I - T^{-\frac{1}{2}}W^\mathsf{T}WT^{-\frac{1}{2}})$ is small, then $\|T^{-\frac{1}{2}}W^\mathsf{T}WT^{-\frac{1}{2}} - I\|_F$ is also small, and $WT^{-\frac{1}{2}}$ is close to the set of orthogonal matrices. Therefore, one interpretation for Theorem 2 is that among all the nonsingular diagonal scaling matrices $T$ satisfying $T - W^\mathsf{T}W \in \mathbf{D}^n$, the choice of $T$ used in AOL makes the scaled weight matrix $WT^{-\frac{1}{2}}$ the closest to the set of orthogonal matrices. This provides a new mathematical explanation of how AOL can generate "almost orthogonal" weights.

One potential issue for AOL is that $\mathbf{D}^n$ is typically much smaller than $\mathbf{S}_+^n$, and the condition $T - W^\mathsf{T}W \in \mathbf{D}^n$ may be too conservative compared to the original condition $T - W^\mathsf{T}W \in \mathbf{S}_+^n$ in Theorem 1. If we denote the set $\hat{\mathbf{T}} = \left\{ T : T \text{ is nonsingular diagonal, and } T - W^\mathsf{T}W \in \mathbf{S}_+^n \right\}$, then we have $\arg\min_{T \in \hat{\mathbf{T}}} \operatorname{tr}(I - T^{-\frac{1}{2}}W^\mathsf{T}WT^{-\frac{1}{2}}) \leq \arg\min_{T \in \mathbf{T}} \operatorname{tr}(I - T^{-\frac{1}{2}}W^\mathsf{T}WT^{-\frac{1}{2}})$, and $\arg\min_{T \in \hat{\mathbf{T}}} \|T^{-\frac{1}{2}}W^\mathsf{T}WT^{-\frac{1}{2}} - I\|_F \leq \arg\min_{T \in \mathbf{T}} \|T^{-\frac{1}{2}}W^\mathsf{T}WT^{-\frac{1}{2}} - I\|_F$. This leads to interesting alternative choices of $T$ which can further promote orthogonality:

$$T = \underset{T \in \hat{\mathbf{T}}}{\arg\min} \|T^{-\frac{1}{2}}W^\mathsf{T}WT^{-\frac{1}{2}} - I\|_F \quad \text{or} \quad T = \underset{T \in \hat{\mathbf{T}}}{\arg\min} \operatorname{tr}(I - T^{-\frac{1}{2}}W^\mathsf{T}WT^{-\frac{1}{2}}) \tag{3}$$

Although (3) may be solved as convex programs on small toy examples, it is not practical to use such choice of $T$ for large-scale problems. It is our hope that our theoretical discussion above will inspire more future research on developing new practical choices of $T$ for promoting orthogonality.

# 5 EXTENSIONS OF CPL: THE POWER OF GERSHGORIN CIRCLE THEOREM

In this section, we extend the original CPL layer (5) to a new family of 1-Lipschitz network structures via providing new analytical solutions to our condition $W^\mathsf{T}W \preceq T$. We term this general family of layers as SDP-based Lipschitz layers (SLL), since the condition $W^\mathsf{T}W \preceq T$ can be viewed as an SDP for the decision variable $T$. First of all, we extend the existing CPL (Eq. (1)) via applying more general choices of $T$ with Theorem 1. From the discussion after Theorem 1, we already know that we can use the choice of $T = \operatorname{diag}(\sum_{j=1}^n |W^\mathsf{T}W|_{ij})$ to replace the original choice $T = \|W\|_2^2 I$. In this section, we will strengthen CPL via an even more general choice of $T$, which is based on a special version of Gershgorin circle theorem. Specifically, we will apply (Horn et al., 2012, Corollary 6.1.6) to show the following result.

**Theorem 3.** *Let $W$ be the weight matrix. Suppose $T$ is a nonsingular diagonal matrix. If there exists some diagonal matrix $Q$ with all positive diagonal entries such that $(T - QW^\mathsf{T}WQ^{-1})$ is a real diagonally dominant matrix with diagonal entries being all positive, then $T \succeq W^\mathsf{T}W$, and the function $h(x) = x - 2WT^{-1}\sigma(W^\mathsf{T}x + b)$ is 1-Lipschitz for $\sigma$ being ReLU,* tanh *or sigmoid.*

We defer the proof of this result to the appendix. If we choose $Q = I$, the above theorem just recovers the choice of $T$ used in AOL, i.e. $T = \operatorname{diag}(\sum_{j=1}^n |W^\mathsf{T}W|_{ij})$. However, it is expected that the use of more general $Q$ will allow us to train a less conservative 1-Lipschitz neural network due to the increasing expressivity brought by these extra variables. We will present numerical results to demonstrate this. We also emphasize that $(T - QW^\mathsf{T}WQ^{-1})$ is typically not a symmetric matrix and hence is not in $\mathbf{D}^n$ even when it only has non-negative eigenvalues. However, this does not affect our proof on the positive-semidefiniteness of $(T - W^\mathsf{T}W)$.

**Application of Theorem 3.** We can parameterize $Q^{-1} = \operatorname{diag}(q_i)$ with $q_i > 0$. Then the $(i,j)$-th entry of $QW^\mathsf{T}WQ^{-1}$ is equal to $(W^\mathsf{T}W)_{ij}q_j/q_i$. Hence we can just set the diagonal entry of $T$ as

$$T_{ii} = \sum_{j=1}^n |(W^\mathsf{T}W)_{ij}q_j/q_i| = \sum_{j=1}^n |W^\mathsf{T}W|_{ij}\frac{q_j}{q_i}. \tag{4}$$

This leads to our new choice of $T = \operatorname{diag}(\sum_{j=1}^n |W^\mathsf{T}W|_{ij}q_j/q_i)$. Notice that the layer function $h(x) = x - 2WT^{-1}\sigma(W^\mathsf{T}x + b)$ has a residual network structure. Hence it is expected that vanishing gradient will not be an issue. Therefore, we can simultaneously optimize the training loss over $W$ and $\{q_i\}$. We will present a numerical study to demonstrate that such a training approach will allow us to generate competitive results on training certifiably robust classifiers.

**SDP conditions for more general network structures.** It is also worth mentioning that the SDP condition in Theorem 1 can be generalized to address the following more general structure:

$$h(x) = Hx + G\sigma(W^\mathsf{T}x + b), \tag{5}$$

where $H$ and $G$ will be determined by the weight $W$ in some manner, and the matrix dimensions are assumed to be compatible. If we choose $H = I$ and $G = -2WT^{-1}$, then (5) reduces to the residual network structure considered in Theorem 1. There are many other choices of $(H, G)$ which can also ensure (5) to be 1-Lipschitz. Our last theoretical result is a new SDP condition which generalizes Theorem 1 and provides a more comprehensive characterization of such choices of $(H, G)$.

**Table 1:** This table presents the natural, provable accuracy as well as the number of parameters and training time of several concurrent work and our SLL networks on CIFAR10 dataset. All results for SLL networks are the result of the average of 3 trainings.

| Models | Natural Accuracy | Provable Accuracy ($\varepsilon$) | | | | Number of Parameters | Time by Epoch (s) |
|---|---|---|---|---|---|---|---|
| | | $\frac{36}{255}$ | $\frac{72}{255}$ | $\frac{108}{255}$ | 1 | | |
| **GloRo** (Leino et al., 2021) | 77.0 | 58.4 | - | - | - | 8M | 6 |
| **Local-Lip-B** (Huang et al., 2021b) | 77.4 | 60.7 | 39.0 | 20.4 | - | 2.3M | 8 |
| **Cayley Large** (Trockman et al., 2021) | 74.6 | 61.4 | 46.4 | 32.1 | - | 21M | 30 |
| **SOC 20** (Singla et al., 2021) | 78.0 | 62.7 | 46.0 | 30.3 | - | 27M | 52 |
| **SOC+ 20** (Singla et al., 2022b) | 76.3 | 62.6 | 48.7 | 36.0 | - | 27M | 52 |
| **CPL XL** (Meunier et al., 2022) | 78.5 | 64.4 | 48.0 | 33.0 | - | 236M | 163 |
| **AOL Large** (Prach et al., 2022) | 71.6 | 64.0 | 56.4 | 49.0 | 23.7 | 136M | 64 |
| **SLL Small** | 71.2 | 62.6 | 53.8 | 45.3 | 20.4 | 41M | 20 |
| **SLL Medium** | 72.2 | 64.3 | 56.0 | 48.3 | 23.9 | 78M | 35 |
| **SLL Large** | 72.7 | 65.0 | 57.3 | 49.7 | 25.4 | 118M | 55 |
| **SLL X-Large** | 73.3 | 65.8 | 58.4 | 51.3 | 27.3 | 236M | 105 |

**Theorem 4.** *Let $n$ be the neuron number. For any non-negative scalars $\{\lambda_i\}_{i=1}^n$, define*

$$\Lambda = \mathrm{diag}(\lambda_1, \lambda_2, \ldots, \lambda_n). \tag{6}$$

*Suppose the activation function $\sigma$ is ReLU or $\tanh$ or sigmoid. If there exist non-negative scalars $\{\lambda_i\}_{i=1}^n$ such that the following matrix inequality holds*

$$\begin{bmatrix} I - H^\mathsf{T}H & -H^\mathsf{T}G - W\Lambda \\ -G^\mathsf{T}H - \Lambda W^\mathsf{T} & 2\Lambda - G^\mathsf{T}G \end{bmatrix} \succeq 0 \tag{7}$$

*then the network layer (5) is 1-Lipschitz, i.e., $\|h(x) - h(y)\| \leq \|x - y\|$ for all $(x, y)$.*

The above theorem can be proved via modifying the argument used in Fazlyab et al. (2019, Theorem 1)[4] and we defer the detailed proof to the appendix. On one hand, if we choose $H = 0$, then our condition (7) reduces to a variant of Theorem 1 in Fazlyab et al. (2019).[5] On the other hand, for residual network structure with $H = I$, we can choose $T = 2\Lambda^{-1}$ and $G = -W\Lambda = -2WT^{-1}$ to reduce (7) to our original algebraic condition $T \succeq W^\mathsf{T}W$. Therefore, Theorem 4 provides a connection between the SDP condition in Fazlyab et al. (2019) and our proposed simple algebraic condition in Theorem 1. It is possible to obtain new 1-Lipschitz network layers via providing new analytical solutions to (7). It is our hope that our proposed SDP condition (7) can lead to many more 1-Lipschitz network structures in the future.

## 6 EXPERIMENTS

In this section, we present a comprehensive set of experiments with 1-Lipschitz neural networks based on our proposed *SDP-based Lipschitz Layer*. More specifically, we build 1-Lipschitz neural networks based on the following layer:

$$h(x) = x - 2W \mathrm{diag}\left(\sum_{j=1}^n |W^\mathsf{T}W|_{ij} q_j / q_i\right)^{-1} \sigma(W^\mathsf{T}x + b), \tag{8}$$

where $W$ is a parameter matrix being either dense or a convolution, $\{q_i\}$ forms a diagonal scaling matrix as described by Theorem 3, and $\sigma(\cdot)$ is the ReLU nonlinearity function. We use the same architectures proposed by Meunier et al. (2022) with *small*, *medium*, *large* and *xlarge* sizes. The architecture consists of several Conv-SLL and Linear-SLL. For CIFAR-100, we use the Last Layer Normalization proposed by Singla et al. (2022b) which improves the certified accuracy when the number of classes becomes large. Note that the layer presented in Equation (8) can be easily implemented with convolutions following the same scaling as in Prach et al. (2022). Our experiments focus on the impact of the Lipschitz layer structures on certified robustness. This complements a recent study on other aspects (e.g. projection pooling) of robust networks (Singla et al., 2022a).

---

[4]As commented in Pauli et al. (2021), such a modification works as long as $\Lambda$ is diagonal.
[5]To see this connection, set $(\alpha, \beta, W^0, W^1) = (0, 1, W^\mathsf{T}, G)$ in Theorem 1 of Fazlyab et al. (2019).

**Table 2:** This table presents the natural and provable accuracy of several concurrent works and our SLL networks on CIFAR100 and TinyImageNet datasets. SLL networks are averaged of 3 trainings.

| Datasets | Models | Natural Accuracy | Provable Accuracy ($\varepsilon$) | | | |
|---|---|---|---|---|---|---|
| | | | $\frac{36}{255}$ | $\frac{72}{255}$ | $\frac{108}{255}$ | 1 |
| CIFAR100 | **Cayley Large** (Trockman et al., 2021) | 43.3 | 29.2 | 18.8 | 11.0 | - |
| | **SOC 20** (Singla et al., 2021) | 48.3 | 34.4 | 22.7 | 14.2 | - |
| | **SOC+ 20** (Singla et al., 2022b) | 47.8 | 34.8 | 23.7 | 15.8 | - |
| | **CPL XL** (Meunier et al., 2022) | 47.8 | 33.4 | 20.9 | 12.6 | - |
| | **AOL Large** (Prach et al., 2022) | 43.7 | 33.7 | 26.3 | 20.7 | 7.8 |
| | **SLL Small** | 44.9 | 34.7 | 26.8 | 20.9 | 8.1 |
| | **SLL Medium** | 46.0 | 35.5 | 27.9 | 22.2 | 9.1 |
| | **SLL Large** | 46.4 | 36.2 | 28.4 | 22.7 | 9.6 |
| | **SLL X-Large** | 46.5 | 36.5 | 29.0 | 23.3 | 10.4 |
| TinyImageNet | **GloRo** (Leino et al., 2021) | 35.5 | 22.4 | - | - | - |
| | **Local-Lip-B (+MaxMin)** (Huang et al., 2021b) | 36.9 | 23.4 | 12.7 | 6.1 | 0.0 |
| | **SLL Small** | 26.6 | 19.5 | 14.2 | 10.4 | 2.9 |
| | **SLL Medium** | 30.4 | 22.3 | 15.9 | 11.6 | 3.0 |
| | **SLL Large** | 31.3 | 23.0 | 16.9 | 12.3 | 3.3 |
| | **SLL X-Large** | 32.1 | 23.2 | 16.8 | 12.0 | 3.2 |

**Details on the architectures & Hyper-parameters.** Table 3 describes the detail of our Small, Medium, Large and X-Large architectures. We trained our networks with a batch size of 256 over 1000 epochs with the data augmentation used by . We use an Adam optimizer (Kingma et al., 2014) with 0.01 learning rate and parameters $\beta_1$ and $\beta_2$ equal to 0.5 and 0.9 respectively and no weight decay. We use a piecewise triangular

**Table 3:** The SLL architecture used for the experiments is inspired by Meunier et al..

| | S | M | L | XL |
|---|---|---|---|---|
| **Conv-SLL** | 20 | 30 | 90 | 120 |
| **Channels** | 45 | 60 | 60 | 70 |
| **Linear-SLL** | 7 | 10 | 15 | 15 |
| **Linear Features** | 2048 | 2048 | 4096 | 4096 |

learning rate scheduler to decay the learning rate during training. We use the CrossEntropy loss as in Prach et al. (2022) with a temperature of 0.25 and an offset value $\frac{3}{2}\sqrt{2}$.

**Results in terms of Natural and Certified Accuracy on CIFAR10/100.** First, we evaluate our networks (SLL) on CIFAR10 and CIAFR100 and compare the results against recent 1-Lipschitz neural network structures: Cayley, SOC, SOC+, CPL and AOL. We also compare SLL with two other Lipschitz training approaches (Leino et al., 2021; Huang et al., 2021b), which do not guarantee prescribed global Lipschitz bounds during the training stage. Table 1 presents the natural and certified accuracy with different radius of certification on CIFAR10. For a fair comparison, parameter number and training time per epoch for each method are also added to Table 1. Results on CIFAR100 are included in Table 2. We can see that our approach outperforms existing 1-Lipschitz architectures including AOL and CPL on certified accuracy for all values of $\varepsilon$. We also observe that SLL-based 1-Lipschitz neural networks offer a good trade-off among previous approaches with respect to natural and certified accuracy. A detailed comparison is given below.

**Advantages of SLL over Cayley/SOC.** In general, it is difficult to compare the expressive power of non-residual and residual networks. Hence we do not claim that with the same model size, SLL is more representative than Cayley or SOC which are not residual networks in the first place. However, we believe that the current choice of $T$ in SLL is very easy to calculate and hence leads to a scalable approach that allows us to train very large models with a reasonable amount of time. For illustrative purposes, consider the comparison between SLL and Cayley in Table 1. We can see that SLL Small has more parameters than Cayley Large (41M vs. 21M) while being faster to train. Indeed, the Cayley approach involves computing an expensive orthogonal projection (with a matrix inverse), while SOC requires to the computation of several convolutions at training and inference (from 6 to 12) to compute the exponential of a convolution up to a desired precision. Hence the training time per epoch for Cayley Large and SOC is actually longer than SLL Small. While being faster to train SLL Small still outperforms Cayley Large and SOC for all three values of $\varepsilon$. In general, we think it is fair to claim that our approach is more scalable than previous approaches based on orthogonal layers, and allows the use of larger networks which leads to improvements in certified robustness.

**Table 5:** The table describes the empirical robustness of our SLL-based classifiers on CIFAR10 ans CIFAR100 datasets. The empirical robustness is measured with *AutoAttacks*. All results are the average of 3 models.

| Models | CIFAR10 – *AutoAttack* ($\varepsilon$) | | | | CIFAR100 – *AutoAttack* ($\varepsilon$) | | | |
|---|---|---|---|---|---|---|---|---|
| | $\frac{36}{255}$ | $\frac{72}{255}$ | $\frac{108}{255}$ | 1 | $\frac{36}{255}$ | $\frac{72}{255}$ | $\frac{108}{255}$ | 1 |
| **SLL Small** | 68.1 | 62.5 | 56.8 | 35.0 | 40.7 | 35.2 | 30.4 | 17.0 |
| **SLL Medium** | 69.1 | 63.8 | 58.4 | 37.0 | 41.5 | 36.4 | 31.5 | 17.9 |
| **SLL Large** | 69.8 | 64.5 | 59.1 | 37.9 | 42.1 | 37.1 | 32.6 | 18.7 |
| **SLL X-Large** | 70.3 | 65.4 | 60.2 | 39.4 | 42.7 | 37.8 | 33.2 | 19.5 |

**Advantages of SLL over AOL/CPL.**  With careful tuning of the offset value, SLL outperforms AOL for all values of $\varepsilon$. We experiment with several offset values: $\sqrt{2}$, $\frac{3}{2}\sqrt{2}$ and $2\sqrt{2}$. The detailed results for all these different offset values are deferred to Table 6 in the appendix. In general, the offset value offers a trade-off between natural accuracy and robustness, thus, by choosing the offset value properly, SLL Large already achieves better results than AOL Large (notice that the training time per epoch for these two is roughly the same). SLL X-Large has even more improvements. We can also see that SLL Large outperforms CPL XL for all values of $\varepsilon$ while being faster to train. For larger value of $\varepsilon$, the gain of SLL over CPL is remarkable (over 10%).

**Results on TinyImageNet.**  We have also implemented SLL on TinyImageNet (see Table 2). Previously, other 1-Lipschitz network structures including SOC, Cayley, AOL, and CPL have not been tested on TinyImageNet, and the state-of-the-art approach on TinyImageNet is the local Lipschitz bound approach (Huang et al., 2021a). We can see that SLL significantly outperforms this local Lipschitz approach for larger values of $\varepsilon$ (while generating similar results for the small $\varepsilon$ case). Notice that the local Lipschitz approach (Huang et al.,

**Table 4:** Inference time for Local-Lip-B and SLL X-Large on the full TinyImageNet validation with 4 GPUs.

| Models | Inference Time |
|---|---|
| **Local-Lip-B** | 41 min |
| **SLL X-Large** | 8 sec |

2021a) is quite different from other 1-Lipschitz network methods in the sense that it has no guarantees on the Lipschitz constant of the resultant network and hence does not generate 1-Lipschtiz networks in the first place. Furthermore, given that this approach does not guarantee a Lipschitz bound during training, a lot more computation needs to be performed during inference, making the certification process very time consuming. Table 4 describes the inference time on TinyImageNet for this local Lipschitz approach and SLL X-large.

**Results on Empirical Robustness.**  We also provide results of our approach on empirical robustness against an ensemble of diverse parameter-free attacks (*i.e.*, *AutoAttacks*) developed by Croce et al. (2020b). Table 5 reports the empirical robustness accuracy for different levels of perturbations. Although *AutoAttacks* is a strong empirical attack consisting of an ensemble of several known attacks: APGD$_{CE}$, APGD$_{DLR}$, FAB (Croce et al., 2020a) and Square (Andriushchenko et al., 2020). We can observe that the measure robustness is high and well above the certified radius. Indeed, on CIFAR10, we observe a robustness "gain" of up to 4.5%, 9.6%, 14.1% and 21.7% for respectively, 36, 72, 108 and 255 $\varepsilon$-perturbations.

## 7  CONCLUSION

In this paper, we present a unifying framework for designing Lipschitz layers. Based on a novel algebraic perspective, we identify a common SDP condition underlying the developments of spectral normalization, orthogonality-based methods, AOL, and CPL. Furthermore, we have shown that AOL and CPL can be re-derived and generalized using our theoretical framework. From this analysis, we introduce a family of SDP-based Lipschitz layers (SLL) that outperforms previous work. In the future, it will be interesting to investigate more expressive structures of $T$ and extending our contributions to address multi-layer neural networks.

ACKNOWLEDGMENTS

This work was performed using HPC resources from GENCI–IDRIS (Grant 2021-AD011013259) and funded by the French National Research Agency (ANR SPEED-20-CE23-0025). A. Havens and B. Hu are generously supported by the NSF award CAREER-2048168.

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

# A  PROOFS

In this section, we present the proofs for the theorems presented in our paper.

## A.1  PROOF OF THEOREM 1

To prove the first statement in Theorem 1, notice that we have

$$\|g(x) - g(y)\|^2 = \|WT^{-\frac{1}{2}}(x - y)\|^2 = (x - y)^\mathsf{T} T^{-\frac{1}{2}} W^\mathsf{T} W T^{-\frac{1}{2}}(x - y).$$

Based on our algebraic condition $W^\mathsf{T} W \preceq T$, we immediately have

$$\|g(x) - g(y)\|^2 \leq (x - y)^\mathsf{T} T^{-\frac{1}{2}} T T^{-\frac{1}{2}}(x - y) = \|x - y\|^2.$$

Therefore, Statement 1 is true.

To prove Statement 2 in Theorem 1, we need to use the property of the nonlinear activation function $\sigma$. Notice that the condition $W^\mathsf{T} W \preceq T$ ensures that all the diagonal entries of the nonsingular matrix $T$ are positive. Therefore, $T^{-1}$ is also a diagonal matrix whose diagonal entries are all positive. For all the three activation functions listed in the above theorem, $\sigma$ is slope-restricted on $[0, 1]$, and the following inequality holds for any $\{x', y'\}$ (Fazlyab et al., 2019, Lemma 1):

$$\begin{bmatrix} x' - y' \\ \sigma(x') - \sigma(y') \end{bmatrix}^\mathsf{T} \begin{bmatrix} 0 & -T^{-1} \\ -T^{-1} & 2T^{-1} \end{bmatrix} \begin{bmatrix} x' - y' \\ \sigma(x') - \sigma(y') \end{bmatrix} \leq 0.$$

We can set $x' = W^\mathsf{T} x + b$ and $y' = W^\mathsf{T} y + b$, and the above inequality becomes

$$\begin{bmatrix} W^\mathsf{T}(x - y) \\ \sigma(W^\mathsf{T} x + b) - \sigma(W^\mathsf{T} y + b) \end{bmatrix}^\mathsf{T} \begin{bmatrix} 0 & -T^{-1} \\ -T^{-1} & 2T^{-1} \end{bmatrix} \begin{bmatrix} W^\mathsf{T}(x - y) \\ \sigma(W^\mathsf{T} x + b) - \sigma(W^\mathsf{T} y + b) \end{bmatrix} \leq 0.$$

We can rewrite the above inequality as

$$\begin{bmatrix} x - y \\ \sigma(W^\mathsf{T} x + b) - \sigma(W^\mathsf{T} y + b) \end{bmatrix}^\mathsf{T} \begin{bmatrix} 0 & -WT^{-1} \\ -T^{-1}W^\mathsf{T} & 2T^{-1} \end{bmatrix} \begin{bmatrix} x - y \\ \sigma(W^\mathsf{T} x + b) - \sigma(W^\mathsf{T} y + b) \end{bmatrix} \leq 0. \tag{9}$$

Now we can apply the following argument:

$$\|h(x) - h(y)\|^2$$
$$= \|x - y - 2\left(WT^{-1}\sigma(W^\mathsf{T} x + b) - WT^{-1}\sigma(W^\mathsf{T} y + b)\right)\|^2$$

$$= \begin{bmatrix} x - y \\ 2WT^{-1}\left(\sigma(W^\mathsf{T} x + b) - \sigma(W^\mathsf{T} y + b)\right) \end{bmatrix} \begin{bmatrix} I & -I \\ -I & I \end{bmatrix} \begin{bmatrix} x - y \\ 2WT^{-1}\left(\sigma(W^\mathsf{T} x + b) - \sigma(W^\mathsf{T} y + b)\right) \end{bmatrix}$$

$$= \begin{bmatrix} x - y \\ \sigma(W^\mathsf{T} x + b) - \sigma(W^\mathsf{T} y + b) \end{bmatrix}^\mathsf{T} \begin{bmatrix} I & -2WT^{-1} \\ -2T^{-1}W^\mathsf{T} & 4T^{-1}W^\mathsf{T}WT^{-1} \end{bmatrix} \begin{bmatrix} x - y \\ \sigma(W^\mathsf{T} x + b) - \sigma(W^\mathsf{T} y + b) \end{bmatrix}$$

$$\leq \begin{bmatrix} x - y \\ \sigma(W^\mathsf{T} x + b) - \sigma(W^\mathsf{T} y + b) \end{bmatrix}^\mathsf{T} \begin{bmatrix} I & -2WT^{-1} \\ -2T^{-1}W^\mathsf{T} & 4T^{-1} \end{bmatrix} \begin{bmatrix} x - y \\ \sigma(W^\mathsf{T} x + b) - \sigma(W^\mathsf{T} y + b) \end{bmatrix},$$

where the last step follows from the fact that our condition $W^\mathsf{T} W \preceq T$ implies $T^{-1}W^\mathsf{T}WT^{-1} \preceq T^{-1}$. Finally, we can combine the above inequality with (9) to show

$$\|h(x) - h(y)\|^2 \leq \begin{bmatrix} x - y \\ \sigma(W^\mathsf{T} x + b) - \sigma(W^\mathsf{T} y + b) \end{bmatrix}^\mathsf{T} \begin{bmatrix} I & 0 \\ 0 & 0 \end{bmatrix} \begin{bmatrix} x - y \\ \sigma(W^\mathsf{T} x + b) - \sigma(W^\mathsf{T} y + b) \end{bmatrix}$$
$$= \|x - y\|^2,$$

which is the desired conclusion. Our proof is complete.

## A.2   Proof of Theorem 2

Since $T$ is nonsingular diagonal and $T - W^\mathsf{T}W \in \mathbf{D}^n$, then we must have $T_{ii} \geq \sum_j |W^\mathsf{T}W|_{ij}$. Given the following key relation:

$$\mathrm{tr}(I - T^{-\frac{1}{2}}W^\mathsf{T}WT^{-\frac{1}{2}}) = \sum_i \left(1 - \frac{|W^\mathsf{T}W|_{ii}}{T_{ii}}\right),$$

it becomes clear that we need to choose the smallest value of $T_{ii}$ for all $i$ to minimize $\mathrm{tr}(I - T^{-\frac{1}{2}}W^\mathsf{T}WT^{-\frac{1}{2}})$. Therefore the choice of $T$ for AOL minimizes $\mathrm{tr}(I - T^{-\frac{1}{2}}W^\mathsf{T}WT^{-\frac{1}{2}})$ over $T \in \mathbf{T}$. The proof for the last equality in Theorem 2 is similar. Let us denote $X = I - T^{-\frac{1}{2}}W^\mathsf{T}WT^{-\frac{1}{2}}$. For any $(i, j)$, the quantity $X_{ij}^2$ is always monotone non-decreasing in $T_{ii}$ and $T_{jj}$. To minimize $\|X\|_F$, we just need to choose the smallest value for all $T_{ii}$ under the constraint $T_{ii} \geq \sum_j |W^\mathsf{T}W|_{ij}$. This completes the proof.   □

## A.3   The Gershgorin Circle Theorem and Proof of Theorem 3

Before stating the proof of Theorem 3, we will state the Gershgorin circle theorem, a useful result from matrix analysis which locates the eigenvalues of a real (or complex) matrix (Horn et al., 2012, Theorem 6.1.1).

**Theorem 5** (Gershgorin). *Let $A \in \mathbb{R}^{n \times n}$ and define the $n$ Gershgorin discs of $A$ by*

$$\left\{ z \in \mathbb{C} : |z - A_{ii}| \leq \sum_{j \neq i} |A_{ij}| \right\}, \quad i \in \{1, \dots, n\}.$$

*Then the eigenvalues of $A$ are contained in the union of Gershgorin discs*

$$\bigcup_{i=1}^{n} \left\{ z \in \mathbb{C} : |z - A_{ii}| \leq \sum_{j \neq i} |A_{ij}| \right\}$$

A useful consequence of this theorem is that whenever $A$ is diagonally dominant (i.e. $|A_{ii}| \geq \sum_{j \neq i} |A_{ij}|$) with positive diagonal entries, then the eigenvalues of $A$ must be non-negative. With this fact, we now proceed to the proof of Theorem 3.

**Proof of Theorem 3**   Given nonsingular matrix $Q$, clearly the eigenvalues of $Q(T - W^\mathsf{T}W)Q^{-1}$ and $(T - W^\mathsf{T}W)$ are the same. If $Q(T - W^\mathsf{T}W)Q^{-1}$ is diagonally dominant and only has positive diagonal entries, then we can apply Gershgorin circle theorem (Horn et al., 2012, Corollary 6.1.6) to show that all the eigenvalues of $Q(T - W^\mathsf{T}W)Q^{-1}$ (which is the same as $T - QW^\mathsf{T}WQ^{-1}$) are non-negative. Therefore, we know that all the eigenvalues of $(T - W^\mathsf{T}W)$ are non-negative. Since $(T - W^\mathsf{T}W)$ is symmetric, we have $T \succeq W^\mathsf{T}W$. Then we can apply Theorem 1 to reach our desired conclusion.   □

## A.4   Proof of Theorem 4

A detailed proof for Theorem 4 is presented here. Our proof is based on modifying the arguments used in (Fazlyab et al., 2019, Theorem 1), and mainly relies on the quadratic constraint technique developed in the control field (Megretski et al., 1997).

First, notice that (7) is equivalent to the following condition:

$$\begin{bmatrix} H^\mathsf{T}H & H^\mathsf{T}G \\ G^\mathsf{T}H & G^\mathsf{T}G \end{bmatrix} \preceq \begin{bmatrix} I & -W\Lambda \\ -\Lambda W^\mathsf{T} & 2\Lambda \end{bmatrix}. \tag{10}$$

Suppose (10) holds. Next we will show that $h(x) = Hx + G\sigma(W^\mathsf{T}x + b)$ is 1-Lipschitz.

For all the three activation functions listed in the above theorem, $\sigma$ is slope-restricted on $[0, 1]$, and the following inequality holds for any $\{x', y'\}$ (Fazlyab et al., 2019, Lemma 1):

$$\begin{bmatrix} x' - y' \\ \sigma(x') - \sigma(y') \end{bmatrix}^\mathsf{T} \begin{bmatrix} 0 & -\Lambda \\ -\Lambda & 2\Lambda \end{bmatrix} \begin{bmatrix} x' - y' \\ \sigma(x') - \sigma(y') \end{bmatrix} \leq 0.$$

**Table 6:** Additional results for CIFAR10 and CIFAR100 datasets with different offset values.

| Offset | Models | CIFAR10 | | | | | CIFAR100 | | | | |
|---|---|---|---|---|---|---|---|---|---|---|---|
| | | Natural Accuracy | Provable Accuracy ($\varepsilon$) | | | | Natural Accuracy | Provable Accuracy ($\varepsilon$) | | | |
| | | | $\frac{36}{255}$ | $\frac{72}{255}$ | $\frac{108}{255}$ | 1 | | $\frac{36}{255}$ | $\frac{72}{255}$ | $\frac{108}{255}$ | 1 |
| $\sqrt{2}$ | SLL small | 73.3 | 63.7 | 53.8 | 44.5 | 15.3 | 46.7 | 35.2 | 26.4 | 20.1 | 5.9 |
| | SLL medium | 74.0 | 64.7 | 54.9 | 45.3 | 16.0 | 47.2 | 36.1 | 27.1 | 20.7 | 6.5 |
| | SLL large | 74.6 | 65.3 | 55.2 | 45.8 | 16.2 | 47.9 | 36.7 | 27.9 | 21.3 | 6.7 |
| | SLL xlarge | 75.3 | 65.7 | 55.8 | 46.1 | 16.3 | 48.3 | 37.2 | 28.3 | 21.8 | 6.9 |
| $\frac{3}{2}\sqrt{2}$ | SLL small | 71.2 | 62.6 | 53.8 | 45.3 | 20.4 | 44.9 | 34.7 | 26.8 | 20.9 | 8.1 |
| | SLL medium | 72.2 | 64.3 | 56.0 | 48.3 | 23.9 | 46.0 | 35.5 | 27.9 | 22.2 | 9.1 |
| | SLL large | 72.7 | 65.0 | 57.3 | 49.7 | 25.4 | 46.4 | 36.2 | 28.4 | 22.7 | 9.6 |
| | SLL xlarge | 73.3 | 65.8 | 58.4 | 51.3 | 27.3 | 46.5 | 36.5 | 29.0 | 23.3 | 10.4 |
| $2\sqrt{2}$ | SLL small | 70.0 | 61.5 | 53.4 | 45.7 | 22.7 | 44.6 | 34.5 | 26.5 | 21.0 | 8.6 |
| | SLL medium | 70.8 | 63.1 | 55.4 | 48.3 | 25.8 | 45.4 | 35.5 | 27.9 | 22.1 | 9.8 |
| | SLL large | 71.4 | 63.9 | 56.7 | 49.8 | 27.8 | 45.9 | 36.0 | 28.2 | 22.7 | 10.3 |
| | SLL xlarge | 71.6 | 64.6 | 57.7 | 50.8 | 29.6 | 46.1 | 36.3 | 29.0 | 23.6 | 11.0 |

We can set $x' = W^\mathsf{T} x + b$ and $y' = W^\mathsf{T} y + b$, and the above inequality becomes

$$\begin{bmatrix} W^\mathsf{T}(x-y) \\ \sigma(W^\mathsf{T}x+b) - \sigma(W^\mathsf{T}y+b) \end{bmatrix}^\mathsf{T} \begin{bmatrix} 0 & -\Lambda \\ -\Lambda & 2\Lambda \end{bmatrix} \begin{bmatrix} W^\mathsf{T}(x-y) \\ \sigma(W^\mathsf{T}x+b) - \sigma(W^\mathsf{T}y+b) \end{bmatrix} \leq 0.$$

We can rewrite the above inequality as

$$\begin{bmatrix} x-y \\ \sigma(W^\mathsf{T}x+b) - \sigma(W^\mathsf{T}y+b) \end{bmatrix}^\mathsf{T} \begin{bmatrix} 0 & -W\Lambda \\ -\Lambda W^\mathsf{T} & 2\Lambda \end{bmatrix} \begin{bmatrix} x-y \\ \sigma(W^\mathsf{T}x+b) - \sigma(W^\mathsf{T}y+b) \end{bmatrix} \leq 0. \quad (11)$$

Now we can apply the following argument:

$$\|h(x) - h(y)\|^2 = \|H(x-y) + \big(G\sigma(W^\mathsf{T}x+b) - G\sigma(W^\mathsf{T}y+b)\big)\|^2$$

$$= \begin{bmatrix} H(x-y) \\ G\big(\sigma(W^\mathsf{T}x+b) - \sigma(W^\mathsf{T}y+b)\big) \end{bmatrix} \begin{bmatrix} I & I \\ I & I \end{bmatrix} \begin{bmatrix} H(x-y) \\ G\big(\sigma(W^\mathsf{T}x+b) - \sigma(W^\mathsf{T}y+b)\big) \end{bmatrix}$$

$$= \begin{bmatrix} x-y \\ \sigma(W^\mathsf{T}x+b) - \sigma(W^\mathsf{T}y+b) \end{bmatrix}^\mathsf{T} \begin{bmatrix} H^\mathsf{T}H & H^\mathsf{T}G \\ G^\mathsf{T}H & G^\mathsf{T}G \end{bmatrix} \begin{bmatrix} x-y \\ \sigma(W^\mathsf{T}x+b) - \sigma(W^\mathsf{T}y+b) \end{bmatrix}$$

$$\leq \begin{bmatrix} x-y \\ \sigma(W^\mathsf{T}x+b) - \sigma(W^\mathsf{T}y+b) \end{bmatrix}^\mathsf{T} \begin{bmatrix} I & -W\Lambda \\ -\Lambda W^\mathsf{T} & 2\Lambda \end{bmatrix} \begin{bmatrix} x-y \\ \sigma(W^\mathsf{T}x+b) - \sigma(W^\mathsf{T}y+b) \end{bmatrix},$$

where the last step follows from the condition (10). Finally, we can combine the above inequality with (11) to show

$$\|h(x) - h(y)\|^2 \leq \begin{bmatrix} x-y \\ \sigma(W^\mathsf{T}x+b) - \sigma(W^\mathsf{T}y+b) \end{bmatrix}^\mathsf{T} \begin{bmatrix} I & 0 \\ 0 & 0 \end{bmatrix} \begin{bmatrix} x-y \\ \sigma(W^\mathsf{T}x+b) - \sigma(W^\mathsf{T}y+b) \end{bmatrix}$$

$$= \|x-y\|^2,$$

which is the desired conclusion.

## B  ADDITIONAL RESULTS

In this section, we will present some additional results and discuss the effect of the offset value on training. The choice of the offset value will affect the performance of SLL significantly. Larger offset values will lead to decrease in natural accuracy and increase in certified robust accuracy. The details are documented in Table 6.

## C FURTHER DISCUSSIONS

In this section, we provide some extra discussions on control-theoretic interpretations and possible extensions of our main results.

### C.1 CONTROL-THEORETIC INTERPRETATIONS FOR OUR MAIN RESULTS

Our work is inspired by the quadratic constraint approach (Megretski et al., 1997) and the Lur'e system theory (Lur'e et al., 1944) developed in the control community. Specifically, the general network layer structure (5) can be viewed as a Lur'e system, which is a feedback interconnection of a linear dynamical system and a static nonlinearity. In this section, we try to make this connection more transparent.

Specifically, we can denote $x' = h(x)$ and rewrite (5) as follows

$$x' = Hx + Gw$$
$$v = W^\mathsf{T}x + b$$
$$w = \sigma(v)$$

which is exactly a shifted version of the Lur'e system. Therefore, it is not surprising that one can tailor the Lur'e system theory to study the properties of (5). As a matter of fact, the previous developments in Fazlyab et al. (2019) and Revay et al. (2020) were based on similar ideas. The main difference is that our paper requires solving SDPs analytically. In the controls literature, the formulated SDP conditions are typically solved numerically.

### C.2 A VARIANT OF THEOREM 1

When discussing AOL and SLL, our main paper makes the assumption that all the columns of $W$ have at least one non-zero entry such that (2) is well defined. To drop this assumption, we can use the following variant of Theorem 1.

**Theorem 6.** *For any weight matrix $W \in \mathbb{R}^{m \times n}$, if there exists a diagonal matrix $\Gamma \in \mathbf{S}^n$ such that $\Gamma W^\mathsf{T} W \Gamma \preceq \Gamma$, then the two following statements hold true.*

1. *The mapping $g(x) = W\Gamma^{\frac{1}{2}}x + b$ is 1-Lipschitz.*

2. *The mapping $h(x) = x - 2W\Gamma\sigma(W^\mathsf{T}x + b)$ is 1-Lipschitz if $\sigma$ is ReLU, $\tanh$ or sigmoid.*

The proof is omitted here, since we can use exactly the same argument as before. If $\Gamma$ happens to be nonsingular, then we can set $T = \Gamma^{-1}$, and the above theorem exactly reduces to Theorem 1. However, the above result allows $\Gamma$ to be singular. This is useful for designing AOL and SLL in the case where $W$ has some zero columns. Suppose the $(i_0, j)$-entry of $W^\mathsf{T}W$ is equal to 0 for all $j$. Then we can set the $(i_0, i_0)$-th entry of $\Gamma$ as 0 and still use (2) or (4) for other entries. It is straightforward to verify that the resultant $\Gamma$ is still a feasible solution to $\Gamma W^\mathsf{T} W \Gamma \preceq \Gamma$, and then we can implement AOL or SLL accordingly.

### C.3 A VARIANT OF THEOREM 3

We can also modify Theorem 3 for the non-residual network layer case. The following variant of Theorem 3 is useful.

**Theorem 7.** *Let $W$ be the weight matrix. Suppose $T$ is a nonsingular diagonal matrix. If there exists some diagonal matrix $Q$ with all positive diagonal entries such that $(T - QW^\mathsf{T}WQ^{-1})$ is a real diagonally dominant matrix with diagonal entries being all positive, then $T \succeq W^\mathsf{T}W$, and the function $g(x) = WT^{-\frac{1}{2}}x + b$ is 1-Lipschitz.*

The proof is trivial and hence omitted. Based on the above result, it is possible that one can use (4) to construct a non-residual layer that can still improve upon AOL.

