# OpenReview forum: "A Unified Algebraic Perspective on Lipschitz Neural Networks"
_ICLR.cc/2023/Conference — ICLR 2023 notable top 25%_

### Official Review · Reviewer_rDGJ · 2022-10-24

**Confidence:** 3
**Correctness:** 4
**Technical Novelty And Significance:** 3
**Empirical Novelty And Significance:** 3
**Recommendation:** 6

**Clarity, Quality, Novelty And Reproducibility:**

The paper is clear and easy to follow, and very well motivated.

One minor point is that it would be useful to write the statement for Gershgorin circle theorem so that the proof for Theorem3 could be easier to follow.

**Strength And Weaknesses:**

[Strengths]

- The paper is well written and easy to follow, and provide a theoretical framework that can unify the existing methodologies to get 1-Lipschitz functions.
- Theorem 1 provides shows that spectral normalization and convex potential layers can be looked at as the linear and residual block for the same value of T.
- The authors apply theorem 3 to come up with a new formulation of T with a new diagonal matrix Q.

[Weakness]

- The empirical verification is lacking. The authors have provided with a general framework for designing Lipschitz Layers, however they only show empirical results with one result from this formulation.

**Summary Of The Paper:**

The paper introduces an SDP based formulation for the development of 1-Lipschitz architecture. They show that many of the existing techniques for generating 1-Lipschitz neural networks can cast as solution to this SGD condition (W^TW < T).

Furthermore, based on the SDP formulation the authors derive new conditions for Lipschitz Layers (based on different values of T) and show through experimentations that the SDP based lipschitz layers get better performance than previous approaches on natural and certified accuracy.

Based on their setup, the paper also provides motivation for why Almost-Orthogonal-Layers promote orthogonality of layers during training.

**Summary Of The Review:**

I think the unifying framework introduced by the authors is interesting. However, the paper lacks enough empirical validation since the authors only show results for one type of layer from their SDP formulation.

---

> ### Author Response · Authors · 2022-11-16
> **Response to Reviewer rDGj**
>
> Thanks for your comments. Here is our response.
>
> **The empirical verification is lacking. The authors have provided with a general framework for designing Lipschitz Layers, however they only show empirical results with one result from this formulation.**
>
> We have compared SLL with all existing methods on CIFAR10/100 and added new results of our method on TinyImageNet. In comparison with existing papers, we believe that our empirical comparison is comprehensive. If the reviewer thinks that there is a certain empirical study which has been performed in other papers on 1-Lipschitz networks but has not been performed by us, please let us know. We will definitely add the simulations if detailed pointers are provided.
>
> In addition, we think the reviewer's criticism on "only show empirical results on one result from this formulation" is unreasonable. From Theorem 1, it is clear that there are infinitely many choices of T which can be used to construct 1-Lipschitz networks. The issue is how to find the "best" choice of T for training deep 1-Lipschitz networks. Based on our empirical study, we find one choice of T which outperforms existing methods for all values of $\epsilon$. We believe it is reasonable for us to focus our report on this particular choice of T in our paper.
> Reporting other choices of T which do not generate the best results seems to be a meaningless thing to do. In our opinion, finding one good choice of T which outperforms existing 1-Lipschitz networks is a solid contribution.
>
> **One minor point is that it would be useful to write the statement for Gershgorin circle theorem so that the proof for Theorem3 could be easier to follow.**
>
> Thanks for this useful suggestion. We have revised our paper accordingly. The statement for Gershgorin circle theorem has been restated before the proof of Theorem 3 (which is now moved to appendix so that more space is made for numerical study).

---

> > ### Comment · Reviewer_rDGJ · 2022-11-23
> > **Thank you for the rebuttal**
> >
> > I thank the authors for their rebuttal. The only reason I said that it would have been nice to see results for some other T is because as the authors claim, there are can be many such Ts and since the claim is that SDP based Lipschitz layers get better performance. I think it is unfortunate that the authors feel that it is unreasonable and meaningless, since I believe that it would only make their proposed method stronger.
> >
> > Thank you for adding extra results on TinyImageNet and also adding the Gershgorin circle theorem statement to the theorem 3 statement. I think the paper is interesting and gives a nice framework.

---

> > > ### Author Response · Authors · 2022-11-24
> > > **Response to Reviewer rDGj**
> > >
> > > Thanks so much for reading our response and providing such useful feedback. We want to follow up with your comments and think this actually provides a great opportunity for us to further clarify several important points in our paper.
> > >
> > > **The only reason I said that it would have been nice to see results for some other T is because as the authors claim, there are can be many such Ts and since the claim is that SDP based Lipschitz layers get better performance. I think it is unfortunate that the authors feel that it is unreasonable and meaningless, since I believe that it would only make their proposed method stronger.**
> > >
> > > Thanks so much for this thoughtful comment which make us realize that we need to make some of our statements more precise. Our true claim in our last response is that "there are infinitely many choices of T which can guarantee the network structure to be 1-Lipschitz." We do not claim that "there are infinitely many choices of T which can achieve SoTA certified robustness." Our true claim is that "with some particular choice of T, SLL gets SoTA certified robustness." Specifically, to achieve good certified robustness, the 1-Lipschitz property has to be combined with the prediction margin. Therefore, it is reasonable to say that there are two separate fundamental questions in the design of Lipschitz networks.
> > >
> > > 1) How to guarantee the networks to be 1-Lipschitz?
> > > 2) Among the many different choices of 1-Lipschitz structures, which one is the best for training networks with SoTA prediction margins and eventually certified robustness?
> > >
> > > Theorem 1 in our paper makes a significant contribution in answering the first question, and shows that there can be infinitely many different 1-Lipschitz network structures based on different T. But Theorem 1 does not address the second question directly. For the second question, we leverage Gershgorin circle theorem to obtain one particular choice of T (which is stated in Theorem 3 in our paper) which eventually leads to SoTA certified robustness on CIFAR10/100 and TinyImageNet. Since we do not claim that there are infinitely many choices of T which can achieve SoTA certified robustness on CIFAR10/100 and TinyImageNet, we think it is reasonable to focus the numerical study of certified robustness on the particular choice of T from Theorem 3 which is currently the best choice we find. In the future, it is possible that one can build upon our current framework and find other choices of  T which achieve better certified robustness than Theorem 3. Actually it is our hope that our current results can serve as a promising initial step which inspire more researchers to work on new choices of T and further push the certified robustness of 1-Lipschitz networks.
> > >
> > > **Thank you for adding extra results on TinyImageNet and also adding the Gershgorin circle theorem statement to the theorem 3 statement. I think the paper is interesting and gives a nice framework.**
> > >
> > > Thanks so much for the positive feedback. We hope that our response above does clarify our true claims on our results. We will revise our paper accordingly to make these points clear if it gets accepted.

---

### Official Review · Reviewer_VQD9 · 2022-10-25

**Confidence:** 4
**Correctness:** 4
**Technical Novelty And Significance:** 3
**Empirical Novelty And Significance:** 2
**Recommendation:** 8

**Clarity, Quality, Novelty And Reproducibility:**

**Clarity**: The paper is written clearly and it is rather easy to follow the paper's discussion.

**Quality**: The paper discusses an interesting unified framework for Lipschitz regularization methods.

**Novelty**: While the paper revisits several existing Lipschitz regularization schemes, the unified framework and discussed extensions look novel to me.

**Reproducibility**: While I have not checked the details of numerical experiments, the numerical evaluation seems reproducible to me.

**Strength And Weaknesses:**

Strengths:

1- The paper has an interesting theoretical contribution and proposes a nice unified framework for Lipschitz regularization methods in deep learning.

2- The paper is well-written and easy to follow. The theorems are sufficiently explained and well supported by several examples.

Weaknesses:

1- The numerical results are somewhat preliminary and discussed for only CIFAR-10 and CIFAR-100 datasets. Discussing the results for one or two large-scale image dataset could provide a stronger support for the SLL approach. Also, the performance scores for the SLL method and AOL baseline are quite close and in several cases, AOL performs better than the proposed SLL apporach.

2- While the theorem proofs are interesting, I think they could have been presented in the Appendix. The proofs' space could have been used to discuss additional numerical results.

**Summary Of The Paper:**

This paper introduces a unified framework for a group of well-known Lipschitz regularization methods in deep learning. Theorem 1 presents the algebraic condition on normalizing matrix $T$ which requires $WW^\top - T\preceq \mathbf{0}$. As a result, the framework includes spectral normalization, orthogonal regularization, Almost-Orthogonal-layer (AOL), Convex Potential Layers (CPL) schemes. Subsequently, Theorem 2 gives a new interpretation of AOL methods by observing that the chosen matrix $T$ results in the closest normalized matrix to the set of orthogonal matrices which is an important step toward achieving a $1$-Lipschitz network. Then, in Section 5 the authors generalize the CPL method by introducing new optimization variables in a diagonal matrix $Q$ which is simultaneously optimized with weight matrix $W$. The paper discusses several numerical results to show the merits of the proposed SLL method on robust learning problems over CIFAR-10 and CIFAR-100 datasets.

**Review Update**

The authors' response satisfactorily addresses my comments and I therefore raise my score to 8.

**Summary Of The Review:**

This paper proposes an interesting unified framework for Lipschitz regularization methods, and the theorems help extend and generalize existing Lipschitz regularization schemes. The numerical contribution could be further improved by experimenting the proposed methods on more datasets and finding factors in which they outperform the baseline AOL.

---

> ### Author Response · Authors · 2022-11-16
> **Response to Reviewer VQD9**
>
> Thanks so much for the useful comments. Detailed responses to your comments are provided below.
>
> **Q:The numerical results are somewhat preliminary and discussed for only CIFAR-10 and CIFAR-100 datasets. Discussing the results for one or two large-scale image dataset could provide a stronger support for the SLL approach. Also, the performance scores for the SLL method and AOL baseline are quite close and in several cases, AOL performs better than the proposed SLL approach.**
>
> A: Thanks so much for this useful comment. We have tuned SLL more carefully and now our results outperform AOL in all cases. In addition, we have also run our approach on TinyImageNet and achieved state-of-the-art certified robustness there. Here are some details.
>
> ### SLL vs. AOL
> First,  we tried several different offset values in training SLL, and observed that the offset value is a hyperparameter that affects the trade-off between natural accuracy and robustness. After tuning this offset value carefully, our approach (SLL) successfully outperforms AOL and all other approaches on certified robust accuracy. We discuss the results of different offset values in details in Table 6 of our revised paper. A summary for the case with the offset value being $\frac32 \sqrt{2}$ is given in Table 2 of our revised paper and restated below.
>
> | Models |  $\frac{36}{255}$  | $\frac{72}{255}$ | $\frac{108}{255}$ | 1 |
> | -----------  | :-----------: | :-----------: | :-----------: | :-----------: |
> | AOL Large    |  64.0 | 56.4 | 49.0 | 23.7 |
> | SLL Large    |  65.0 | 57.3 | 49.7 | 25.4 |
> | SLL X-Large  |  65.8 | 58.4 | 51.3 | 27.3 |
>
> With careful tuning of the offset value (choosing it to be $\frac32 \sqrt{2}$), SLL Large already outperforms AOL Large for all values of $\varepsilon$ (the training time per epoch for these two are roughly the same, i.e. 55s for SSL Large vs. 64s for AOL Large). SLL X-Large has even more improvements.
>
> ### TinyImageNet: SLL vs. "Local-Lip-B+MaxMin"
> We have also added results for SLL on TinyImageNet (see Table 2 in our revised paper). Previously, other 1-Lipschitz network structures including SOC, Cayley, AOL, and CPL have not been tested on TinyImageNet, and the state-of-the-art approach on TinyImageNet is the local Lipschitz bound approach (Local-Lip-B+MaxMin) from (Huang et al 2021). We have the following comparison.
>
> | Models | $\frac{36}{255}$  | $\frac{72}{255}$ | $\frac{108}{255}$ | 1 |
> | -----------  | :-----------: | :-----------: | :-----------: | :-----------: |
> | Local-Lip-B+ MaxMin |  23.4 | 12.7 |  6.1 | 0.0 |
> | SLL Large    |  23.0 | 16.9 | 12.3 | 3.3 |
> | SLL X-Large  | 23.2 | 16.8 | 12.0 | 3.2 |
>
>
> We can see that SLL significantly outperforms "Local-Lip-B+MaxMin" for larger values of $\epsilon$ (while generating similar results for the small $\varepsilon$ case). Notice that the local Lipschitz approach is quite different from other 1-Lipschitz network methods in the sense that it has no guarantees on the Lipschitz constant of the resultant network and hence does not generate 1-Lipschtiz networks in the first place. Given that this approach does not guarantee a Lipschitz bound during training, a lot more computation needs to be performed during inference, making the certification process very time consuming. The comparison for inference time in the certification stage is listed here. See Table 4 in our revised paper for more details.
>
> | Models      | Inference Time |
> | ----------- | :-----------: |
> | Local-Lip-B | 41 mins       |
> | SLL X-Large   | 8 sec        |
>
>  In comparison with the local Lipschitz approach, SLL has the following advantages:
> 1. SLL guarantees the global Lipschitz constant to be 1 while the local Lipchitz approach does not have any guarantees on either global or local Lipschitz constants.
> 2. In the certification step, the local Lipschitz approach requires significant computation efforts for evaluating local Lipschitz bound for points in the testing sets while our approach does not require such computation by using a known guaranteed global Lipschitz constant (which is 1).
> 3. On TinyImageNet, SLL significantly outperforms this local Lipschitz approach for larger values of $\epsilon$ (while generating very similar results for the small $\epsilon$ case).
> 4. For CIFAR10 and CIFAR100, SLL significantly outperforms this local Lipchitz approach for all values of $\epsilon$.
>
> Overall, we think the advantages of our proposed approach have been convincingly established via our extensive numerical study.  Especially for relatively large $\epsilon$ values, the gains of our approach are significant over all existing approaches.
>
> **While the theorem proofs are interesting, I think they could have been presented in the Appendix. The proofs' space could have been used to discuss additional numerical results**
>
> Thanks. We have moved all the proofs to the appendix, and use the space to include numerical results on TinyImageNet and detailed comparisons between our approach and several other methods.

---

> > ### Comment · Reviewer_VQD9 · 2022-11-20
> > **Thank you for your response**
> >
> > I thank the authors for their thoughtful responses and for using my suggestions to further improve the numerical section. Based on the response and revised paper, I raise my score to 8.

---

### Official Review · Reviewer_PHyU · 2022-10-25

**Confidence:** 3
**Correctness:** 4
**Technical Novelty And Significance:** 4
**Empirical Novelty And Significance:** 4
**Recommendation:** 8

**Clarity, Quality, Novelty And Reproducibility:**

This paper is well written, and the proposed perspective and techniques are novel.
However, the template of the paper seem to be wrong and should be fixed.

**Strength And Weaknesses:**

Strength:
1) The authors provide a unified perspective on the development of 1-Lipschitz layers, which provides better understanding of the connections between different methods, and allows for new parameterizations of 1-Lipschitz layer.
2) Theorem 1 can recover a wide range of previous methods on parameterization 1-Lipschitz layers, and leads to varaitions of new parameterizations.
3) The authors show some computational efficient parameterizations as an extension to CPL. The performance on certified robustness surpasses previous 1-Lipschitz neural networks and further reduces the gap between certified accuracy and AutoAttack accuracy, implying the new parameterization is more expressive.

Weaknesses:
1) The architectures that are used for SLL seem to be much larger than those used in OrthoConv (Trockman et al., 2021). For instance, Cayley-Large uses an architecture with 4 convolutional layers and 3 fully connected layers, while in Table 3, SLL-small has 20 convolutional layers and 7 fully connected layers. It may be good to also put number of parameters in the table performance comparison table, and comment on the effectiveness of different architectures of 1-Lipschitz neural network.
2) What activation functions do the authors use for SLL? It seems that the authors follow the architecture choice in Meunier et al. (2022) but it'd be good to mention the activation function choice because it is important for training 1-Lipschitz neural networks. Some previous works (e.g. Trockman et al., 2021) finds that gradient-norm-preserving GroupSort is much more effective than ReLU in training 1-Lipschitz neural networks. I am curious on whether this is also the case for the residual architecture.

**Summary Of The Paper:**

This paper proposes a unified perspective on 1-Lipschitz layers. The 1-Lipschitz layers in previous literature can be recovered by the algebraic conditions in Theorem 1. The theoretical results also lead to new parameterizations of 1-Lipschitz neural network layers. Experiments on certified robustness shows the effectiveness of the new paramterizations.

**Summary Of The Review:**

This paper proposes a unified perspective on the development of 1-Lipschitz neural network layer, and provides new parameterizations of 1-Lipschitz layer. The experiments show strong certified robustness results which demonstrate the flexibility of the new parameterization.

---

> ### Author Response · Authors · 2022-11-16
> **Response to Reviewer PHyU**
>
> We thank the reviewer for the positive feedback. We have fixed the template issue.  Detailed responses to your comments are provided below.
>
>
> **Q: The architectures that are used for SLL seem to be much larger than those used in OrthoConv (Trockman et al., 2021). For instance, Cayley-Large uses an architecture with 4 convolutional layers and 3 fully connected layers, while in Table 3, SLL-small has 20 convolutional layers and 7 fully connected layers. It may be good to also put number of parameters in the table performance comparison table, and comment on the effectiveness of different architectures of 1-Lipschitz neural network.**
>
> A: Thanks for this useful feedback. We have added the number of parameters (network size) of all the methods to Table 1 in our revised paper. We believe that training time per epoch is another important metric, and we also include this information in our table. In general, it is difficult to compare the expressive power of non-residual (SOC, Cayley) and residual networks (CPL, SLL). Hence we do not claim that with the same model size, SLL is more representative than Cayley or SOC which are not residual networks in the first place. However, the current choice of T is very easy to calculate, and hence leads to a scalable approach that allows us to train very large models with reasonable amount of time. For illustrative purposes,  the comparison between our SLL approach and the previous Cayley approach on CIFAR10 is summarized as follows
>
> | Models      | $\frac{36}{255}$  | $\frac{72}{255}$ | $\frac{108}{255}$ | Params | sec/epoch |
> | ----------- | :---------------: | :--------------: | :---------------: | :---------------: | :---------------: |
> | Cayley Large |   61.4 | 46.4 | 32.1 | 21M | 30 |
> | SLL Small  |  62.6 | 53.8 | 45.3 | 41M | 20 |
>
> We can see that SLL Small has more parameters than Cayley Large (41M vs. 21M) while being faster to train. Indeed, the Cayley approach involves computing an expensive orthogonal projection (with a matrix inverse), while SOC requires to computes several convolution at training and inference (from 6 to 12) to compute the exponential of a convolution up to a desired precision. Hence the training time per epoch for Cayley Large and SOC is actually longer than SLL Small (see Table 1 in our revised paper). While being faster to train SLL Small still outperforms Cayley Large and SOC for all three values of $\epsilon$. In general, we think it is fair to claim that our approach is more scalable than previous approaches based on orthogonal layers, and allows the use of larger networks which lead to the improvements in certified robustness.
>
>
> **What activation functions do the authors use for SLL? It seems that the authors follow the architecture choice in Meunier et al. (2022) but it'd be good to mention the activation function choice because it is important for training 1-Lipschitz neural networks. Some previous works (e.g. Trockman et al., 2021) finds that gradient-norm-preserving GroupSort is much more effective than ReLU in training 1-Lipschitz neural networks. I am curious on whether this is also the case for the residual architecture.**
>
> We use ReLU for SLL. Our theory for residual networks (Statement 2 in Theorem 1) requires the activation function to be slope-restricted on [0,1]. This assumption does not hold for GroupSort (which is only slope-restricted on [-1,1]), and hence replacing ReLU with GroupSort in SLL may not guarantee 1-Lipschitzness any more. Intuitively, the residual structure helps overcome the gradient vanishing issue, and hence the GNP property may not be that crucial for training 1-Lipschitz residual networks. We also agree with the reviewer that it will be interesting to see whether there are new activation functions which can significantly outperform ReLU in the residual network setting. We leave this as a future task.

---

### Official Review · Reviewer_BFUj · 2022-10-29

**Confidence:** 3
**Correctness:** 4
**Technical Novelty And Significance:** 3
**Empirical Novelty And Significance:** 3
**Recommendation:** 8

**Clarity, Quality, Novelty And Reproducibility:**

The paper is overall well written and has good novelty. The paper provides good insights on how to design 1-Lipschitz neural networks. Sufficient details on experiments are given in appendix.

**Strength And Weaknesses:**

Strengths:

1. The new formulation for representing the 1-Lipschitz layer is novel. The theoretical analysis provide many insights on how to design neural network layers that are one Lipschitz.

2. Empirical results look very promising - the proposed method achieves better provable robustness (one main application of 1-Lipschitz network) than prior works consistently, although in some cases the improvements are marginal.


Weaknesses:

1. One question is not clearly answered by this paper: Is it possible to show that the formulation proposed in this paper can universally approximate any 1-Lipschitz functions? If it is hard to prove theoretically, we hope to know if the new factorization for 1-Lipschitz neural network is more representative compared to previous ones. The paper provides some experimental comparisons, however some baselines use models of different sizes. Ideally, for a fair comparison the number of optimizable parameters should be same in each method.

2. Table 1 should include other recent baselines for provable robustness for L2 norm, such as (Leino et al., 2021) and (Huan et al., 2021); it should be easy to add the results directly. In addition, I think it is important to report training time of each method.

References:

[1] Leino, Klas, Zifan Wang, and Matt Fredrikson. "Globally-robust neural networks." International Conference on Machine Learning. PMLR, 2021.
[2] Huang, Yujia, et al. "Training certifiably robust neural networks with efficient local lipschitz bounds." Advances in Neural Information Processing Systems 34 (2021): 22745-22757.


**Summary Of The Paper:**

This paper proposes a new formulation for designing 1-Lipschitz neural networks, which is more general than existing ones. To guarantee 1-Lipschitzness, existing works use a variety of parameterizations. This paper proposes a more general parameterization which covers existing ones (existing works can be seen as special cases of this parameterization), and the new parameterization allows more flexibility on designing the network. Additionally, experimental results on CIFAR-10 and CIFAR-100 show that the 1-Lipschitz network trained using the new parameterization performs better compared to baselines.

**Summary Of The Review:**

Training 1-Lipschitz neural network is an important and challenging problem and I like the new formulation in this paper because it provides new insights to this problem. The empirical results also look positive. So I tend to accept this paper. I hope the authors can address the weaknesses questions above.

---

> ### Author Response · Authors · 2022-11-16
> **Response to Reviewer BFUj**
>
> We thank the reviewer for acknowledging our contributions. Detailed responses to your comments are provided below.
>
> **Q: One question is not clearly answered by this paper: Is it possible to show that the formulation proposed in this paper can universally approximate any 1-Lipschitz functions? If it is hard to prove theoretically, we hope to know if the new factorization for 1-Lipschitz neural network is more representative compared to previous ones. The paper provides some experimental comparisons, however some baselines use models of different sizes. Ideally, for a fair comparison the number of optimizable parameters should be same in each method.**
>
> A: We thank the reviewer for this useful comment. We have not been able to prove the approximation power of our proposed network structure theoretically. That is definitely an interesting future direction. We also do not claim that with the same model size, the current choice of T leads to more representative networks than previous ones. However, the current choice of T is very easy to calculate, and hence leads to a scalable approach that allows us to train very large models with reasonable amount of time.  For illustrative purpose, first consider the comparison between our SLL approach and the previous Cayley approach.  The comparison on CIFAR10 is summarized as follows
>
> | Models      | $\frac{36}{255}$  | $\frac{72}{255}$ | $\frac{108}{255}$ | Params | sec/epoch |
> | ----------- | :---------------: | :--------------: | :---------------: | :---------------: | :---------------: |
> | Cayley Large |   61.4 | 46.4 | 32.1 | 21M | 30 |
> | SLL Small  |  62.6 | 53.8 | 45.3 | 41M | 20 |
>
> We can see that SLL Small has more parameters than Cayley Large (41M vs. 21M) while being faster to train. Indeed, the Cayley approach involves computing an expensive orthogonal projection (with a matrix inverse), while SOC requires to computes several convolution at training and inference (from 6 to 12) to compute the exponential of a convolution up to a desired precision. Hence the training time per epoch for Cayley Large and SOC is actually longer than SLL Small (see Table 1 in our revised paper). While being faster to train SLL Small still outperforms Cayley Large and SOC for all three values of $\epsilon$. In general, we think it is fair to claim that our approach is more scalable than previous approaches based on orthogonal layers, and allows the use of larger networks which lead to the improvements in certified robustness.
>
>
> Next, we compare SLL with CPL, which is a more recent work and outperforms previous ones.
> We make the following comparison:
> | Models | $\frac{36}{255}$  | $\frac{72}{255}$ | $\frac{108}{255}$ | # of parameters | sec/epoch |
> | ----------- | :-----------: | :-----------: | :-----------: |:-----------: | :---------------: |
> | SLL Small    |   62.6 | 53.8 | 45.3 | 41M |  20 |
> | CPL Small    |   62.3 | 46.9 | 32.2 | 41M |  21 |
> | SLL Medium   |   64.3  |  56.0 |  48.3  | 78M |  35 |
> | CPL Medium   |    63.3 |   47.5  |   32.5 | 78M |  40 |
> | SLL Large    |  65.0 | 57.3 | 49.7  | 118M |  55 |
> | CPL Large    |   63.9 | 48.1 | 32.9 | 118M |  93 |
> | SLL X-Large  | 65.8 | 58.4 | 51.3 |  236M | 105 |
> | CPL X-Large  |  64.4  | 48.0  | 33.0 | 236M | 163 |
>
>
> We can see that with the same size, the training time for SLL and CPL are roughly on the same order, and hence in this case it makes sense to directly compare their performance with a fixed size. With the same size, SLL outperforms CPL for all values of $\epsilon$.  For larger value of $\epsilon$, the gain of SLL over CPL is remarkable (over 10%). On CIFAR100, a similar observation can be made.
>
> **Q: Table 1 should include other recent baselines for provable robustness for L2 norm, such as (Leino et al., 2021) and (Huan et al., 2021); it should be easy to add the results directly. In addition, I think it is important to report training time of each method.**
>
> A: Thank you for bringing these papers to our attention. We have added both to our related work and added the results to our table for comparison.  Our method outperforms these methods for all three values of $\epsilon$. We also added the training time of our models and all the other methods as well as the number of parameters (network size) to Table 1 of our revised paper. These information is used for making a fair comparison between our approach and other methods. Please see Table 1 in our revised paper and related discussions.

---

### Author Response · Authors · 2022-11-16
**General Response**

We sincerely thank the reviewers for their detailed and constructive comments. We provide below a general response to some common concerns and separate replies to each review. We have made changes in the paper (and generated a revised version) in line with these responses.

### Adding network size and training time to the main paper \& fair comparison
We have added parameter number and training time per epoch to Table 1 for fair comparison. Please see Table 1 in our revised paper.

### Comparison with SOC and Cayley
In general, it is difficult to compare the expressive power of non-residual and residual networks. Hence we do not claim that with the same model size, SLL is more representative than Cayley or SOC which are not residual networks in the first place. However, we believe that the current choice of T in SLL is very easy to calculate, and hence leads to a scalable approach that allows us to train very large models with reasonable amount of time. For illustrative purposes, consider the comparison between SLL and Cayley.
| Models      | $\frac{36}{255}$  | $\frac{72}{255}$ | $\frac{108}{255}$ | Params | sec/epoch |
| ----------- | :---------------: | :--------------: | :---------------: | :---------------: | :---------------: |
| Cayley Large |   61.4 | 46.4 | 32.1 | 21M | 30 |
| SLL Small  |  62.6 | 53.8 | 45.3 | 41M | 20 |

We can see that SLL Small has more parameters than Cayley Large (41M vs. 21M) while being faster to train.
Indeed, the Cayley approach involves computing an expensive orthogonal projection (with a matrix inverse), while SOC requires to computes several convolution at training and inference (from 6 to 12) to compute the exponential of a convolution up to a desired precision.
Hence the training time per epoch for Cayley Large and SOC is actually longer than SLL Small.
While being faster to train SLL Small still outperforms Cayley Large and SOC for all three values of $\varepsilon$.
In general, we think it is fair to claim that our approach is more scalable than previous approaches based on orthogonal layers, and allows the use of larger networks which lead to the improvements in certified robustness.


### Comparison with AOL and CPL
  In our original submission, the performance scores for the SLL method and AOL baseline are quite close and in several cases, AOL performs better than the proposed SLL approach. In revision, we have tried several different offset values in training SLL, and observed that the offset value is a hyperparameter that affects the trade-off between natural accuracy and robustness.  With careful tuning of the offset value, SLL outperforms AOL for all values of $\varepsilon$.
We experiment with several offset values: $\sqrt{2}$, $\frac32 \sqrt{2}$ and $2\sqrt{2}$. The detailed results for all these different offset values are given in Table 6 in the appendix of our revised paper. By choosing the offset value properly, SLL Large already achieves better results than AOL Large (notice that the training time per epoch for these two are roughly the same).
| Models | Natural | $\frac{36}{255}$  | $\frac{72}{255}$ | $\frac{108}{255}$ | 1 |
| ----------- | :-----------: | :-----------: | :-----------: | :-----------: | :-----------: |
| AOL Large    | 71.6 | 64.0 | 56.4 | 49.0 | 23.7 |
| SLL Large    | 72.7 | 65.0 | 57.3 | 49.7 | 25.4 |
| SLL X-Large  | 73.3 | 65.8 | 58.4 | 51.3 | 27.3 |

SLL X-Large has even more improvements. We can also see that SLL Large outperforms CPL XL for all values of $\varepsilon$ while being faster to train.  We add these discussions into Section 6 of our revised paper.


### TinyImageNet
In the previous work, other 1-Lipschitz network structures including SOC, Cayley, AOL, and CPL have not been tested on TinyImageNet due to the scalability issue, and the SoTA approach on TinyImageNet is the local Lipschitz approach from (Huang et al., 2021) ("Local-Lip-B+MaxMin").  We tested our approach (SLL) on TinyImageNet and the comparison with Local-Lip-B+MaxMin is provided below (see Table 2 in revision).

| Models |$\frac{36}{255}$  | $\frac{72}{255}$ | $\frac{108}{255}$ | 1 |
| ----------- |  :-----------: | :-----------: | :-----------: | :-----------: |
| Local-Lip-B +MaxMin |  23.4 | 12.7 |  6.1 | 0.0 |
| SLL Medium   |  22.3 | 15.9 | 11.6 | 3.0 |
| SLL Large    |  23.0 | 16.9 | 12.3 | 3.3 |
| SLL X-Large  |  23.2 | 16.8 | 12.0 | 3.2 |

Our method outperforms this local Lipschitz approach for larger values of $\epsilon$ (while generating similar results for the small $\epsilon$ case). The local Lipschitz approach is quite different from other 1-Lipschitz network methods, since it has no guarantees on the Lipschitz constant of the resultant network and does not generate 1-Lipschtiz networks in the first place. Given that this approach does not guarantee a Lipschitz bound during training, a lot more computation needs to be performed during inference, making the certification process time consuming (see Table 4 in our revised paper).

---

### Decision · Program_Chairs · 2023-01-20

**Decision:**

Accept: notable-top-25%

**Justification For Why Not Higher Score:**

This work may meet the bar for an oral presentation, depending on the strength of other contenders.

**Justification For Why Not Lower Score:**

This is a non-obvious unifying framework which captures a variety of existing methods, and it leads to a novel Lipschitz-constrained architecture which outperforms existing approaches. It strikes me as a solid paper overall, and will probably be in the top quarter of papers at the conference.

**Metareview: Summary, Strengths And Weaknesses:**

The contributions include introducing a unifying framework for various methods for training Lipschitz-constrained neural networks (useful for adversarial robustness, control systems, etc.) and using this framework to discover a novel and non-obvious Lipschitz-constrained architecture. Impressively, the general framework can easily handle convolution layers, which have often required complicated constraints. Experiments show improved results on provable adversarial robustness compared with a variety of prior approaches.

The reviewers are uniformly enthusiastic about this paper, and appreciate the generality of the approach and the clarity of the exposition. Some reviewers had concerns about experimental details (e.g. choice of architecture for the baselines), but the authors addressed these concerns with a variety of new experiments. Overall, I think this is a solid paper and recommend acceptance.

**Note From Pc:**

if the above contains the word "oral" or "spotlight" please see: "oral" presentation means -> notable-top-5% and "spotlight" means -> notable-top-25%. As stated in our emails, we are disassociating presentation type from AC recommendations